# Hybrid Core-Shell Polymer Scaffold for Bone Tissue Regeneration

**DOI:** 10.3390/ijms23094533

**Published:** 2022-04-20

**Authors:** Luciana Sartore, Chiara Pasini, Stefano Pandini, Kamol Dey, Marco Ferrari, Stefano Taboni, Harley H. L. Chan, Jason Townson, Sowmya Viswanathan, Smitha Mathews, Ralph W. Gilbert, Jonathan C. Irish, Federica Re, Piero Nicolai, Domenico Russo

**Affiliations:** 1Department of Mechanical and Industrial Engineering, University of Brescia, 25133 Brescia, Italy; c.pasini012@unibs.it (C.P.); stefano.pandini@unibs.it (S.P.); kamolacct@gmail.com (K.D.); 2Department of Applied Chemistry and Chemical Engineering, Faculty of Science, University of Chittagong, Chittagong 4331, Bangladesh; 3Section of Otorhinolaryngology—Head and Neck Surgery, Department of Neurosciences, University of Padua—“Azienda Ospedale Università di Padova”, 35129 Padua, Italy; marco.ferrari@unipd.it (M.F.); stefanotaboni@gmail.com (S.T.); piero.nicolai@unipd.it (P.N.); 4Guided Therapeutics (GTx) Program International Scholar, University Health Network (UHN), Toronto, ON M5G 2A2, Canada; 5Guided Therapeutics (GTx) Program, Techna Institute, University Health Network, Toronto, ON M5G 2A2, Canada; harley.chan@rmp.uhn.ca (H.H.L.C.); jason.townson@rmp.uhn.ca (J.T.); jonathan.irish@uhn.ca (J.C.I.); 6Osteoarthritis Program, Schroeder Arthritis Institute, Krembil Research Institute, Institute of Biomedical Engineering, University Health Network, University of Toronto, Toronto, ON M5G 2A2, Canada; sowmya.viswanathan@uhnresearch.ca (S.V.); smitha.mathews@uhnresearch.ca (S.M.); 7Princess Margaret Cancer Centre, Department of Otolaryngology—Head and Neck Surgery/Surgical Oncology, University Health Network, Toronto, ON M5G 2A2, Canada; ralph.gilbert@uhn.ca; 8Bone Marrow Transplant Unit, ASST Spedali Civili, Department of Clinical and Experimental Sciences, University of Brescia, 25123 Brescia, Italy; federica.re@unibs.it (F.R.); domenico.russo@unibs.it (D.R.)

**Keywords:** tissue engineering, bone tissue regeneration, hydrogels, open-pore PLA-PCL core, hybrid polymer device, composite scaffolds, human mesenchymal stromal cells

## Abstract

A great promise for tissue engineering is represented by scaffolds that host stem cells during proliferation and differentiation and simultaneously replace damaged tissue while maintaining the main vital functions. In this paper, a novel process was adopted to develop composite scaffolds with a core-shell structure for bone tissue regeneration, in which the core has the main function of temporary mechanical support, and the shell enhances biocompatibility and provides bioactive properties. An interconnected porous core was safely obtained, avoiding solvents or other chemical issues, by blending poly(lactic acid), poly(ε-caprolactone) and leachable superabsorbent polymer particles. After particle leaching in water, the core was grafted with a gelatin/chitosan hydrogel shell to create a cell-friendly bioactive environment within its pores. The physicochemical, morphological, and mechanical characterization of the hybrid structure and of its component materials was carried out by means of infrared spectroscopy, thermogravimetric analysis, scanning electron microscopy, and mechanical testing under different loading conditions. These hybrid polymer devices were found to closely mimic both the morphology and the stiffness of bones. In addition, in vitro studies showed that the core-shell scaffolds are efficiently seeded by human mesenchymal stromal cells, which remain viable, proliferate, and are capable of differentiating towards the osteogenic phenotype if adequately stimulated.

## 1. Introduction

Polymeric biomaterials are of key importance in the production of biocompatible and bioresorbable scaffolds for tissue engineering. With the aim of developing scaffolds with enhanced bioactive response, actively promoting new tissue formation, many hydrogels have been studied with keen interest. Hydrogels offer numerous advantages including biocompatibility, biodegradability, fabrication versatility, tunable properties, water absorption comparable to biological tissues and excellent physicochemical mimicry of the natural extracellular matrix (ECM) [1,2,3,4,5]. Of particular interest are hybrid hydrogels that combine the benefits of natural and synthetic components. Naturally derived hydrogels are inherently biocompatible and bioactive, promoting many cellular functions, whereas synthetic ones show better physical properties, controllable degradation rate, and microstructure [6,7,8]. Several studies highlighted the bioactive role of hybrid hydrogels in promoting cell growth, especially when chitosan, a natural polysaccharide, was added to the network [9]. In our recent study we demonstrated that gelatin/chitosan scaffolds displayed faster and higher stress relaxation, which can improve cell spreading, proliferation and differentiation [10]. In the perspective of bone and cartilage regeneration, these hydrogels were found to induce osteogenic differentiation [11] and support chondrogenic differentiation [12].

It is known that biomaterial-based scaffolds act both as bioactive substrate, providing topographical and chemical stimuli for cell proliferation and differentiation, and as structural support during cell growth and ECM deposition [13]. The scaffold must have sufficient mechanical properties to temporarily substitute the missing tissue and to permit essential physiological functions. The reconstruction of head and neck bony defects following tumor ablation or trauma is a typical example of the challenges faced in surgical reconstruction, needing to address multiple issues, including mechanical support, physiological function, and restoration of morphology and aesthetics [14]. Despite the many advantages of hydrogels, their physico-mechanical properties are typically inadequate for the temporary replacement of some specific tissues, particularly in the case of mineralized tissues such as bone. Therefore, bone tissue engineering can receive a great advantage from stiffer synthetic biopolymers, which provide mechanical stability as well as an easy tailoring of the scaffold properties by properly selecting the polymer from the wide variety available [15,16]. Several materials have been thoroughly studied and modified with the intent of finding the most suitable for bone tissue bioengineering, better matching the properties of the host tissue. Among all, some synthetic polyesters, such as poly(lactic acid) (PLA), poly(glycolic acid) (PGA), poly(ε-caprolactone) (PCL) and their combinations as blends and copolymers, are of great interest because of their biocompatibility, biodegradability, convenient processing to produce ECM-like scaffolds with a high degree of complexity, and good mechanical properties. In addition, they are already approved in several commercial forms and formulations for clinical applications such as resorbable sutures and fixation devices [17,18].

In earlier studies, we developed a novel composite material based on PLA matrix and crosslinked particles of sodium polyacrylate (SAP), to obtain superabsorbent thermoplastic products by using melt blending [19,20]. This system was proven to produce highly interconnected porous materials, owing to the ability of SAP particles to swell and to be more easily leached into water. In this way, the interconnected porosity was achieved bio-safely, avoiding solvents or other chemical issues. Moreover, this structure closely mimics both the morphology and the stiffness of bones, and it proved suitable for cell proliferation, especially when PLA was combined with PCL. Biomimicry is a basic concept of tissue bioengineering, consisting of the imitation of the biological architecture, the chemical composition and/or the mechanical behavior of natural tissues [21]. In particular, biomimetic scaffolds for bone tissue engineering have been studied with interest because they can support and accelerate the formation of new tissue similar to the native one [22]. Therefore, these PLA-PCL porous materials turned out to be very promising as scaffolds for bone regeneration when spontaneous healing fails because of the excessive loss of tissue.

The advantages of bioactive materials and stiffer synthetic biopolymers can be combined by the realization of hybrid scaffolds, for which several design strategies and compositions were investigated to address different biological tissues [23]. In some works, bioactive properties were specifically introduced on the surface of the scaffolds, employing surface modification of aliphatic polyesters in order to tune their biomimetic behavior [24]. In many other cases, hydrogels were reinforced by incorporating various types of structured materials, such as particles, fibers, and lattice structures, or by using interpenetrating/semi-interpenetrating networks [25]. In recent studies, hybrid scaffolds were developed with a composite “core-shell structure”, a solution that allows the achievement of a wider variety of properties [26,27,28]. In fact, component types and proportions can be designed to modulate mechanical properties and degradability for specific tissue regeneration, also for obtaining a timing of resorption/degradation synchronous with that of new tissue formation.

In this paper, a 3D integrated core-shell structure was prepared by combining a stiffer thermoplastic core of PLA-PCL for mechanical support and a softer bioactive hydrogel (Hy) shell for the promotion of cell proliferation and differentiation. The two components also have different degradation timing. The thermoplastic is typically resorbed within 1–2 years [29], therefore retaining mechanical stability for long times, whereas the hydrogel maintains its integrity up to 1–2 months [10], rapidly creating space for the growth of new bone tissue. Its morphology was studied by means of a scanning electron microscope (SEM) with the aim of deepening the knowledge about the derived porous structure. To better understand the properties of the materials in their porous state and the effect of the hydrogel presence, thermal and mechanical analyses were carried out both before and after hydrogel grafting. The core-shell structure turned out similar to the spongy bone not only in terms of morphology, but also in terms of stiffness. Furthermore, in order to investigate its possible application as scaffold for tissue engineering, its biological properties were evaluated by in vitro studies investigating vitality, proliferation, and differentiation of human bone marrow-derived mesenchymal stromal cells (hBM-MSCs).

## 2. Results

### 2.1. Preparation and Physicochemical Properties

A novel procedure was developed for the preparation of hybrid scaffolds, consisting of the following steps (Figure 1).

PLA and PCL were mixed by melt blending, which allowed us to obtain a homogeneous product, and SAP particles were then added to the blend. Considering the affine chemical nature of the crosslinked polymeric particles with the PLA-PCL blend, it was observed that easy particle dispersion was obtained by melt mixing. The composite PLA-PCL-SAP blend was then compression molded to make plates and the final shapes of specimens and scaffolds were die-cut from molded plates.Specimens and scaffolds were immersed in water, where SAP particles swelled and leached out of the matrix to form a PLA-PCL core with an interconnected open porosity.A hydrogel-forming solution was prepared and grafted onto the PLA-PCL core to produce the hybrid scaffold (PLA-PCL-Hy), which was then subjected to freeze-drying. A final post-curing treatment was performed in order to complete the grafting/crosslinking of the hydrogel shell.

Before characterization, the materials were exhaustively washed to eliminate any possible unreacted soluble products.

Fourier-transform-infrared (FTIR) analysis was carried out on the core-shell sample, namely PLA-PCL-Hy, and on samples of its components to confirm the qualitative chemical composition of the scaffold. Indeed, the infrared spectra (Figure 1) showed the signals of the presence of all the materials constituents. Many similarities may be noticed in PLA-PCL and PLA-PCL-Hy spectra, both displaying the typical absorbance peaks of the ester group, associated to the polyesters in the core, PLA and PCL (C=O stretching: 1750 cm^−1^ for PLA, 1730 cm^−1^ for PCL; C-O stretching: several peaks between 1000 cm^−1^ and 1300 cm^−1^ for both the constituents). These peaks are partially summed and overlapped, but without shifting; this suggests that no change in the material functional groups and molecular structure has occurred. The neat hydrogel reveals, among other things, the typical absorption peaks around 1630 cm^−1^ and 1544 cm^−1^, respectively, ascribed to amide I (C=O stretching vibration) and amide II (-NH bending and C–N stretching vibrations). These reflexes may be found also in the PLA-PCL-Hy spectrum and are the only difference with respect to the PLA-PCL spectrum, indicating that the hydrogel was successfully incorporated and is coating the core structure.

Figure 2 reports the results of thermogravimetric analyses (TGA) of PLA-PCL before and after hydrogel grafting and of the neat hydrogel. The tests highlighted the thermal stability of porous PLA-PCL up to 300 °C, which was maintained also after the grafting process with the hydrogel. Indeed, after the first mass reduction, due to moisture elimination, both core and core-shell systems show an important mass reduction at about 300 °C, ascribed to the degradation of PLA and PCL in the core. In the case of PLA-PCL-Hy, the main drop is followed by a minor process; in fact, as confirmed by TGA on neat hydrogel, about half of the degradation of the hydrogel is concentrated between 280 °C and 450 °C, but the remaining part occurs only at higher temperatures. Similar results were obtained on samples taken at different distances from the surface of the specimens, confirming that the hydrogel has penetrated deep inside the core and that its content is consistent with the expected composition, i.e., about 2–3% by weight (Table 1).

The water absorption of the materials was evaluated (Table 1) by immersion in water for 24 h. The water uptake was of about 152% (±18%) and 150% (±30%) for PLA-PCL and PLA-PCL-Hy, respectively. There is no statistical difference between these values, probably because the low amount of hydrogel does not significantly contribute to the amount of water trapped inside the pores.

### 2.2. Morphological Characterization

Figure 3 shows SEM images of cryogenically obtained cross-sections for the PLA-PCL-SAP biocomposite material, for porous PLA-PCL and for the core-shell assembly, PLA-PCL-Hy. The biocomposite material (Figure 3A,D) shows a uniform distribution of SAP particles with sharp edges, particle size within a range of 20–50 µM and a weak adhesion to the matrix. During the water treatment, the SAP particles, probably due to the low interaction with the matrix, swell and leach out from the matrix generating high porosity. As demonstrated in a previous study [20], particles leaching was completed after immersion in water for 15 days. Indeed, the elemental analysis carried out during the SEM observation before the water treatment confirmed the identification of SAP particles. On the other hand, after the water treatment, the same analysis carried out in different points of the sample did not detect the presence of the SAP components, suggesting the correct removal of the SAP particles. In addition, it is also possible to notice smaller spherical particles with particle size ranging from 5 to 10 µM. The presence of such spherical domains can be observed also on the pore walls of the porous core (Figure 3B,E) and of the core-shell scaffold (Figure 3C,F), where SAP particles are replaced by a number of interconnected pores. These spherical particles were attributed to PCL domains confirming the immiscibility of the PLA-PCL blend [30,31].

The micrographs reveal that the pores are heterogeneous with regard to size, but homogeneously distributed throughout the cross-section and well interconnected. In addition, the high magnification micrographs (Figure 3E,F) also show more in detail that the inner part of the material is characterized by an interconnected open porous structure, which is conducive to the infiltration of cells. The cellular structure of the core-shell assembly is similar to that of the core, except for a barely visible smooth phase covering the pores, associated to the hydrogel shell. Furthermore, due to the low amount of grafted hydrogel and the high surface extension of the porous core, it is not possible to highlight the thickness of the coating. The hydrogel shell probably covers the surface of the whole core, including the pores, with a very thin layer that represents a favorable environment for cells. Based on the evaluation of pore diameters on images taken with the optical microscope, the pore size ranges between 70 and 150 µM in the case of the core-shell structure, with an average diameter of about 100 ± 20 µM; the same results were obtained by measuring pore size on the neat core specimen (pore size within a range of 60–140 µM; average diameter: 90 ± 20 µM), suggesting that the presence of the hydrogel does not alter the pore size.

### 2.3. Mechanical Properties

Stiffness and strength of the prepared materials were investigated under various loading conditions in order to ensure adequate deformability and structural integrity of the scaffolds and to provide a wider description of their mechanical response in consideration of the complex state of stress to which bones are subjected. The tests were always carried out in wet conditions, i.e., after having immersed the samples in distilled water at body temperature (37 °C).

The results of tensile, compression, and flexure tests are reported as stress vs. strain curves in Figure 4A–C, respectively, comparing the response of PLA-PCL and PLA-PCL-Hy specimens; the curves here displayed are the most representative for each material and testing condition. The results are also reported in terms of Young’s modulus and failure stress and strain in Table 2.

Tensile tests revealed for both materials the attainment of a maximum point, which is here taken as representative of material strength and of the maximum strain that may be applied, before mechanical failure starts to take place, as testified by following a progressive reduction of material properties. Scattering of the results, as that shown in Table 2, was expected due to the heterogeneous cellular structure of the specimens. PLA-PCL and PLA-PCL-Hy show similar values of stiffness (around 100 MPa), whereas the core-shell has an average higher strength and ductility with respect to the neat core structure.

Uniaxial compression tests were carried out to measure stiffness and strength under compressive conditions, particularly important in the case of bones and possibly critical in the case of porous materials. Compression curves revealed a similar trend for the two materials, showing a maximum point at large strain values (about 10%), consisting in local structural collapsing of the cells. When subjected to compression, PLA-PCL-Hy and PLA-PCL exhibit similar strength, but the former shows a slightly higher stiffness than the latter (70 ± 15 MPa vs. 50 ± 17 MPa).

Finally, flexure tests were carried out in order to study another important loading condition for bones, with the specific aim to evaluate flexural stiffness and possible failure conditions. The results of these tests are displayed here as stress vs. strain curves (σ_f,max_ vs. ε_f,max_), instead of the more common load vs. displacement curves; such a representation was adopted to easily compare flexural results with those obtained under tension and compression. Moreover, these curves display a stress peak, followed by a progressive reduction of material properties until failure. The ultimate flexural parameters (i.e., failure stress and strain) were evaluated in correspondence of this peak. Furthermore, it was observed that failure occurred in the central portion of the specimens, producing damage under compression in close proximity to the bending nose and propagating in different directions throughout the thickness; however, the specimens never broke in two halves. Once again, a toughening effect was observed in the coated scaffolds, showing higher values of failure stress and failure strain, whereas their flexural stiffness was found to be the same as that of the uncoated ones (around 200 MPa).

### 2.4. Results of In Vitro Viability, Proliferation, and Osteogenic Differentiation Assay

In vitro tests were performed to assess the vitality, proliferation and differentiation of hBM-MSCs seeded on PLA-PCL-Hy, compared to neat hydrogel samples. Figure 5A,B shows epifluorescence microscopy of the materials seeded with 1000 cell/mm^3^ hBM-MSCs and stained with calcein (green, live cells) and propidium iodide (red, dead cells). A good cell vitality was observed both onto the neat hydrogel (Figure 5A) and PLA-PCL-Hy (Figure 5B). Furthermore, epifluorescence images proved that the distribution of cells in the PLA-PCL-Hy system was not limited to the seeding surface, rather cells migrate and populate the whole thickness of the scaffold. To demonstrate that the hBM-MSCs were capable of osteogenic differentiation on the hydrogels, we performed a standard three-week osteogenic differentiation assay. Figure 5C–F reports epifluorescence microscopy of the neat hydrogel and PLA-PCL-Hy seeded with 2000 cell/mm^3^ hBM-MSCs, on which cells were capable of proliferating (Figure 5C,D) and differentiating towards the osteogenic phenotype (Figure 5E,F), as evidenced by positive staining for osteocalcin, a mature osteoblast bone formation biomarker. Overall, these epifluorescence images show that the employed materials are capable of hosting viable mesenchymal stromal cells that can proliferate and acquire an osteogenic phenotype.

## 3. Discussion

Regeneration of bone tissue over the relatively short distances generally found in bone fractures occurs easily in most healthy patients. Bone injuries beyond simple fractures, however, present greater therapeutic challenges and remain a difficult clinical problem. In the field of head and neck oncologic surgery, ablations resulting in large bony defects mandate immediate reconstruction to maintain vital functions such as chewing, swallowing, and to obtain good cosmetic contour and appearance. Current strategies to reconstruct bony defects require free tissue transfer with possible flap failure, donor site morbidity, risk of plate exposure following radiotherapy, and other risks of prolonged surgery [14]. Therefore, there remains an unmet need for techniques and materials that are effective in promoting bone growth and that attain the increasing complexity and the more and more demanding performances required by biomedical engineering [32]. Innovations in material design and manufacturing processes, indeed, are raising the possibility of producing implants with good performance. Our study aims at developing a multicomponent biocompatible and bioresorbable scaffold that offers physiologically balanced porosity for positive cell exchange and vascular infiltration while providing temporary structural support. To obtain the desired characteristics, we combined a rigid thermoplastic polymer, which forms the core of the scaffold, and a soft grafted hydrogel, which forms its shell. The rigid core provides a mechanical support compliant with the growing bone tissue, whereas the role of the hydrogel is to promote cell colonization and differentiation and to enhance the integration of the scaffold into the hosting tissue. An extensive interconnected porosity, both in the rigid core, to incorporate the gel, and in the final core-shell structure, to ensure full colonization and vascularization, completed the scaffold design.

To surpass some regulatory hurdles, we selected information-rich natural and synthetic polymers, extensively investigated for tissue engineering. Here, we exploited the synergistic contribution of hard and soft polymers as well as natural and synthetic polymers, adopting a novel synthesis approach that consisted of three sequential steps: Figure 1A preparing an appropriate biocomposite material containing leachable particles, Figure 1B forming a porous core by particle swelling and leaching and Figure 1C grafting a hydrogel onto the core to produce a structurally stable micro–macroporous core-shell scaffold. The biocomposite material was based on a binary system containing crosslinked sodium polyacrylate particles, commonly used as superabsorbent polymer, and a PLA-PCL blend that has shown very promising characteristics for scaffold production in tissue engineering. According to our previous work [19], cross-linked sodium polyacrylate particles retain their superabsorbent capacity even when distributed in a thermoplastic matrix. In the presence of water, the biphasic polymer system shows excellent swelling properties, the particles swell and leave the PLA-PCL matrix generating a highly porous core. It was demonstrated that a PLA-PCL composite having 30% SAP particles produces a macroporous network with interconnected porosity of about 60% void volume. This particle content is excellent both for the obtained porosity and for limiting possible microcracks, caused by the swelling of the SAP particles before leaching from the matrix, thus preserving the integrity and mechanical performance of the final porous structure [20,33]. The core was produced with a highly porous structure to enable cells penetration into the whole volume of the final core-shell scaffold. The interconnected porosity is also fundamental for gases, nutrients and metabolic wastes exchange. The last step of the preparation involved the grafting of a hydrogel layer onto the surface of the porous PLA-PCL core to regulate the interactions between cells and host material. The porous core was immersed in a hydrogel-forming solution that was forced under vacuum to penetrate deeply inside the pores, to obtain in situ simultaneously grafting and crosslinking of the component. A biocompatible hydrogel was selected to enable both chondrogenic and osteogenic differentiation of human bone marrow mesenchymal cells. The hydrogel was composed of natural macromolecules, namely chitosan and gelatin, as well as synthetic poly(ethylene glycol) diglycidyl ether used as a crosslinking agent [10]. In fact, it is known that the high reactivity of epoxides exploits the possibility of reacting not only with primary amino groups of gelatin and chitosan, but also with different functionals groups in both acidic and basic conditions. Grafting/crosslinking processes occurred, favoring the formation of the hydrogel on the porous surface of the core, finally followed by washing to remove the soluble and eventually unreacted components. Ultimately, the synthetic procedure aims to improve biocompatibility and bioactivity by modifying the surface of the scaffold, including the pores, in order to promote its osteointegration.

FTIR analysis qualitatively confirmed the composition of the core-shell material by highlighting the presence of all the relevant peaks associated with its components, included those related to amide I and amide II, typical of the hydrogel molecular structure (Figure 1). Furthermore, the material composition was explored by means of TGA tests (Figure 2), that confirmed that the hydrogel penetrated throughout the whole core specimen and was effectively deposited, for an amount estimated to be around 2–3 wt%, as expected. Moreover, the hydrogel grafting did not alter the thermal stability of the PLA-PCL core.

The water uptake of PLA-PCL-Hy is close to that of PLA-PCL, suggesting that the hydrogel coating, despite its hydrophilic behavior (neat hydrogel may swell up to 700% in water [10]), does not increase the water absorption of the final composite material. This is mainly due to the small amount of hydrogel in the core-shell assembly (2–3% by weight) and it is likely that the possible volume increase of the hydrogel upon swelling may be limited anyway by the constraint represented by the pore size of the core. On the other hand, the water uptake observed in the core despite the hydrophobic nature of the PLA-PCL blend, is likely due to water remaining entrapped inside the small pores of the core. Morphological analysis, performed by optical and electronic microscopy on cryogenically obtained cross-sections of the dried samples, allowed us to observe in detail the structure of PLA-PCL-SAP, PLA-PCL and PLA-PCL-Hy. The weak adhesion of SAP particles to the matrix, observed in Figure 3A,D, facilitated the creation of pores by particle leaching. Pictures of the porous PLA-PCL core (Figure 3B,E) and of the core-shell assembly (Figure 3C,F) display a homogeneous distribution of interconnected pores, resembling that of spongy bone tissue and playing a fundamental role in the promotion of cell colonization. On average, the pores are bigger than SAP particles, because the superabsorbent polymer swells before leaching out of the matrix, slightly deforming and fracturing the PLA-PCL matrix, which helps to create further interconnection between the pores and makes their walls moderately rugged. Moreover, the low amount of grafted hydrogel did not alter the distribution of the pores in the core nor their dimensions, which were in line with the size requirement to ensure cell life and diffusion, i.e., pore size of at least about 100 µM [34,35].

The core-shell scaffolds can be compared with spongy bone tissue not only because of their interconnected porous structure, but also in terms of stiffness (spongy bone Young’s modulus: 30–2000 MPa; spongy bone compressive strength: 0.3–50 MPa [36,37]). These results are particularly interesting in the perspective of bone regeneration by tissue engineering, because matrix stiffness is known to profoundly influence cell function and fate [13,38,39]. In addition, for all the tested conditions, the hydrogel shell does not significantly alter the core stiffness, and failure stress and failure strain tend to assume higher values in PLA-PCL-Hy specimens compared to PLA-PCL ones (Table 2). Such a toughening effect may be due to a contribution of the hydrogel resulting in milder local collapse of the core structural cells. In fact, the hydrogel seems to have an adhesive effect that reduces the consequence of any microfractures produced by the swelling and leaching of the SAP particles.

In the end, this hybridization approach allowed us to obtain an outstanding improvement of mechanical properties in comparison to neat gelatin/chitosan scaffolds (stiffness and strength are more than two orders of magnitude higher than those of neat hydrogel [10]), while maintaining the advantage of surface biocompatibility and bioactivity thanks to the addition of the shell layer to the core.

In vitro analyses (Figure 5) showed promising results in terms of capability of PLA-PCL-Hy scaffolds to sustain cell viability, proliferation, and osteogenic differentiation if adequately stimulated. In fact, cells colonized not only the surface but the whole thickness of the sample, as demonstrated by epifluorescence images. Moreover, the material selected for the shell of the scaffolds revealed to be a promising substrate for bone tissue engineering, since proliferation and differentiation assays detected actively proliferating cells and osteocalcin-producing cells on the hydrogel.

Preliminary in vivo studies on mandibular reconstruction in a rabbit model [40] also found the scaffolds to be biocompatible, as they were well tolerated without any anti-inflammatory treatment. No signs of surgical site infection were observed after implantation. Besides, the scaffolds were well integrated with the bone tissue surrounding the mandibular defect as well as with the new growing tissue, which was found to gradually replace them, as observed during ex vivo histochemical analyses.

## 4. Materials and Methods

### 4.1. Materials

Poly(L-lactic acid) (PLA) with a nominal weight average molecular weight (Mw) of 199,590 Da was purchased from Nature Works (Blair, NE) under the brand name Ingeo 2002D. The material was dried at 70 °C under vacuum for 12 h before use.

Poly(ε-caprolactone) (PCL) with a number average molecular weight (Mn) of 10,000 Da was supplied by Sigma-Aldrich Co. (Milan, Italy) and used as received.

Cross-linked sodium polyacrylate, a superabsorbent polymer (SAP), was purchased from Evonik Industries AG (Essen, Germany) and had the brand name Produkt T 5066 F; its particle size was lower than 60 µM and its density was 0.7 g/cm^3^. It was dried at 60 °C under vacuum for 12 h before use.

Type A gelatin, G (pharmaceutical grade, 280 bloom, viscosity 4.30 mP, was kindly provided by Italgel (Cuneo, Italy). Poly(ethylene glycol) diglycidyl ether, (PEGDGE) (molecular weight 526 Da) was supplied by Sigma-Aldrich Co. (Milan, Italy). Chitosan, (CH) (molecular weight between 50,000 and 190,000 Da and degree of deacetylation 75–85%) was obtained from Sigma-Aldrich Co (Milan, Italy). Ethylene diamine (EDA), ethanol, and acetic acid were provided by Fluka (Milan, Italy).

All materials were used without further purification.

### 4.2. Core-Shell Scaffold Preparation

Core-shell systems were obtained by separate preparation of a porous PLA-PLC blend for the core and a hydrogel solution for the shell and grafting them with a proper relative ratio.

#### 4.2.1. PLA-PCL Porous Core Preparation

The starting material for the core system was obtained by the melt blending process by using dry starting components through a discontinuous mixer (Brabender, Plastograph, Duisberg, Germany). PLA (45.6 g, 80 wt%) and PCL (11.4 g 20 wt%) were mixed at 180 °C with a screw speed of 50 rpm for a period of 6 min. The obtained PLA-PCL blend (39 g, 68.5 wt%) and SAP (18 g, 31.5 wt%) were then treated in the discontinuous mixer at the same temperature and screw speed for 6 min, in order to adequately distribute the particles throughout the matrix.

The composite material, recovered from the mixing chamber, was dried in an oven under vacuum at 50 °C for 24 h before being shaped as sheet by means of a laboratory compression molding machine (Collin P200 E); an optimized temperature (180 °C), pressure (30 atm) and time program (1 min) was applied. Two hundred bars (length: 5–40 mm; cross-section: 4 × 3 mm^2^) of blended composite material were cut by mechanical saw and then immersed in distilled water and kept at room temperature for 7–10 days, in order to promote swelling and complete leaching out of the SAP particles, finally leading to a porous rigid structure.

#### 4.2.2. Hydrogel Shell Preparation

The biocompatibility and bioactivity of the rigid core were enhanced by the integration with a hydrogel shell composed, among other things, of natural products such as gelatin and chitosan displaying a chemical nature closer to that of the ECM.

The hydrogel was prepared as concentrated aqueous solution, starting from gelatin (G), chitosan (CH), and poly(ethylene glycol) diglycidyl ether (PEGDGE). Gelatin (6 g) was completely dissolved in 65 mL of distilled water at 40 °C under mild magnetic stirring, then functionalized PEG (1.4 g) was added dropwise into the mixture, followed by the addition of CH solution (2 wt% in acetic acid 1%, 32.5 g) and EDA (70 mg).

The reaction mixture was gently, magnetically stirred at 40 °C for 15 min so to ensure grafting and crosslinking, occurring as natural condensation reactions between the amino-groups of gelatin and chitosan and the epoxy groups of PEGDGE, without catalysts, solvents or additives. G, PEG and CH content in the dry sample was 74.3, 17.6, and 8.1 wt%, respectively.

#### 4.2.3. Core-Shell Integration

The PLA-PCL porous samples were immersed in the concentrated hydrogel solution and subjected to three successive void and nitrogen cycles at 40 °C, to fully eliminate entrapped air from the pores and substituting it with the hydrogel solution. The hydrogel grafted samples were then extracted, frozen in liquid nitrogen, and freeze-dried in a lyophilizer. The dried samples were post-cured in oven at 45 °C under vacuum for 3 hto complete the cross-linking reaction of the hydrogel. The hydrogel grafted samples were washed several times with distilled water to eventually remove the unreacted reagents and soluble components and finally freeze-dried in a lyophilizer. The final amount of grafted hydrogel was evaluated at 3% by weight.

The dry samples were packed into polypropylene bags and sealed under vacuum. Packed samples were sterilized by gamma irradiation with Cobalt 60 gamma rays (dose: 27–33 kGy, according to UNI EN ISO 11,137—Sterilization of Health Care Products). The final composition of PLA-PCL and PLA-PCL-Hy is reported in Table 1.

### 4.3. Physicochemical Characterization

The morphological analysis of the prepared materials was explored by means of optical and electron microscopy on cryogenically fractured cross-sections of dry samples. The optical one employed is a reflected light digital microscope (Leica DMS 300). The pore size was evaluated in terms of pore diameter, as evaluated on 100 pores by means of an image analysis software (Image J). SEM analysis was carried out by means of a LEO EVO 40 scanning electron microscope. The samples were cryogenically fractured in transverse direction in order to better reveal the inner structure of the PLA-PCL-SAP, of the PLA-PCL porous blend and of the PLA-PCL-Hy grafted product. Samples were mounted with carbon tape on aluminum stubs and then sputter coated with gold to make them conductive prior to SEM observation.

Fourier-transform-infrared (FTIR) analysis was carried out on dry samples by means of a Thermo Scientific, Nicolet iS50 FTIR spectrophotometer (Thermo Fisher Scientific, Madison, WI, USA) equipped with a PIKE MIRacle attenuated total reflectance attachment. The spectra were recorded over a range of 400–4000 cm^−1^ at a resolution of 4 cm^−1^. In order to evaluate the qualitative chemical composition of the materials before and after the hydrogel grafting, the test was carried out on PLA-PCL-Hy, on PLA-PCL and on starting PLA, PCL, and Hy.

Thermogravimetric analysis (TGA) was performed on dry samples by means of a TGA Q500 (TA Instruments New Castle, DE, USA), along a heating ramp at 50 °C/min from 30 °C to 800 °C. In addition, samples of neat hydrogel were tested following the same protocol.

Furthermore, water absorption tests were carried out in order to evaluate the structure ability to take up water and fluids thanks to the porosity and hydrogel coating. The specimen weight was evaluated by means of a laboratory balance (Gibertini E42-B) both in wet condition (obtained by a 24 h immersion in distilled water at 37 °C) and in dry condition. The amount of absorbed water was evaluated as follows:Absorbed water [%] = [(m_wet_ − m_dry_)/m_dry_] × 100,(1)
where m_dry_ and m_wet_ are the specimen weight before and after immersion, respectively.

### 4.4. Mechanical Characterization

Mechanical tests were performed under different loading conditions (tensile, three-point bending, compression) by means of an electromechanical dynamometer (Instron, Mod. 3366) with a 500 N load cell. The tests were carried out on porous PLA-PCL and PLA-PCL-Hy specimens after 24 h immersion in distilled water at 37 °C, on at least four specimens per testing condition. The dimensions of each specimen type were measured at the microscope (Leica DMS 300).

#### 4.4.1. Tensile Tests

Uniaxial tensile tests were carried out to measure stiffness, strength and failure strain, and to provide a representation of the material stress vs. strain response. The tests were performed at room temperature with a crosshead speed of 0.5 mm/min; the specimens had a cross section of about 4 × 3 mm^2^, an overall length equal to 40 mm and a gauge length equal to 10–15 mm (according to the conditions that ensured the best gripping without any risk of failure within the grips).

The tensile stress, σ_t_, was evaluated as a nominal value by dividing the measured force, F, for the outer initial cross section, A_0_, of the specimen
σ_t_ = F/A_0_.(2)

The tensile strain, ε_t_, was evaluated as percentage ratio between the crosshead displacement, Δl, and the initial gauge length, l_0_, of the specimen
ε_t_ [%] = (Δl/l_0_) × 100.(3)

The results were represented in terms of stress vs. strain curves, on which the following parameters where calculated: (i) tensile Young’s modulus, as initial slope of the stress vs. strain curve; (ii) tensile failure stress, as maximum stress before specimen break; (iii) tensile failure strain, as strain in correspondence to the maximum stress.

#### 4.4.2. Compression Tests

Uniaxial compression tests were carried out on specimens having a height of about 8 mm and a cross section of about 4 × 3 mm^2^, cut from longer bars in order to identify the region with the most regular cross-section and ensure the most parallel condition between the specimen faces and the compression plates. The specimens were compressed at room temperature between flat and parallel plates, with a crosshead speed of 10 mm/min, up to failure or, in case this did not occur, up to a maximum strain equal to 30%.

The compressive stress, σ_c_, was evaluated as a nominal value by dividing the measured force, F, for the outer initial cross-section, A_0_, of the specimen
σ_c_ = F/A_0_.(4)

The compressive strain, ε_c_, was evaluated as percentage ratio between the crosshead displacement, Δl, and the initial height, h_0_, of the specimen
ε_c_ [%] = (Δl/l_c_) × 100.(5)

The results were represented in terms of stress vs. strain curves, on which the following parameters where calculated: (i) compressive modulus, as initial slope of the stress vs. strain curve; (ii) compressive failure stress, evaluated as peak stress on the stress vs. strain curve.

#### 4.4.3. 3-Point Bending Tests

Flexure tests were carried out at room temperature, adopting a 3-point bending configuration and a crosshead speed of 1 mm/min, on specimens with an overall length equal to 40 mm, a support span equal to 30 mm, and a cross section with width equal to 4 mm and thickness equal to 3 mm.

Flexural stress, σ_f,max_, and strain, ε_f,max_, were evaluated as referred to outer fibers at the midpoint (i.e., the point subjected to maximum stress and strain values in the specimen) by using the following Equations (6) and (7)
σ_f,max_ = 3F·span/2w·th^2^;(6)
ε_f,max_ [%] = 6δ·th/span^2^ × 100,(7)
where F represents the force, δ the maximum deflection of the center of the specimen, w and th the specimen width and thickness, respectively.

The results were represented in terms of stress vs. strain curves, on which the following parameters were calculated: (i) flexural modulus, as initial slope of the stress vs. strain curve; (ii) flexural failure stress, as peak stress on the stress vs. strain curve; (iii) flexural failure strain, as strain in correspondence to the peak stress.

### 4.5. In Vitro Biological Characterization

Scaffolds made of PLA-PCL-Hy and neat hydrogel measuring 5 × 4 × 3 mm^3^ were processed by immersion in a 5-mL solution of “Dulbecco’s Modified Eagle Medium” (DMEM, Sigma-Aldrich Co, Milan, Italy) and 5% human platelet lysate (hPL) for 24 h in standard cell culture condition (37 °C, CO_2_ 5%). The immersion of the scaffold in the growth medium containing human bone marrow-derived mesenchymal stromal cells (donated from healthy consenting donors under an approved protocol in the Viswanathan lab) was considered as time 0. The growth medium was renewed every 24 h in sterile conditions. The scaffolds were tested through 3 different in vitro experiments including epifluorescence microscopy. Images were acquired on an AxioZoom microscope (Zeiss, Jena, Germany) with Plan NeoFluar Z 1× objective NA 0.25. Illumination was from an X-Cite 120 metal halide lamp. Imaging was acquired by using a Hamamatsu ORCA Flash v2 sCMOS camera. Images were deconvolved by using Huygens Professional (Scientific Volume Imaging) and deconvolved by using a Classic Maximum Likelihood Estimation (CMLE) algorithm, with 25 iterations. Deconvolved images were analyzed by using Imaris (Bitplane Software, Belfast, UK, a Division of Oxford Imaging). The following assays were performed.

In vitro vitality assay: the scaffolds were removed from the growth medium, stained with calcein (Invitrogen—Thermo Fisher Scientific, Waltham, MA, USA; green, live cells) and propidium iodide (Bioshop; red, dead cells), and scanned with a 2-channel epifluorescence microscope (red, green). The analysis was performed on day 4 and day 11 in PLA-PCL-Hy and hydrogel scaffolds seeded with 1000 cell/mm^3^ hBM-MSCs.In vitro proliferation assay: the scaffolds were removed from the growth medium, stained with anti-Ki67 rabbit polyclonal antibodies (Abcam; red, actively proliferating cells), 4′,6-diamidino 2-phenylindole (DAPI) (Roche–Millipore Sigma, Sigma-Aldrich Co, Milan, Italy); blue, nuclei), and phalloidin (Invitrogen–Thermo Fisher Scientific, Waltham, MA, USA; green, cytoplasm) and scanned with a 3-channel epifluorescence microscope (red, green, blue). The analysis was performed on day 4 and day 11 in neat hydrogel scaffolds seeded with 2000 cell/mm^3^ hBM-MSCs.In vitro osteogenic differentiation assay: at day 4, the DMEM-hPL medium was substituted with osteogenic medium (DMEM-low glucose with L-glutamine and sodium bicarbonate; hPL 5%; L-Ascorbic acid 50 mg/mL; β-glycerophosphate 10 mM; dexamethasone 10–8 mM) and renewed every 48 h until day 21 (or day 25 post-seeding). Thereon, scaffolds were stained with anti-osteocalcin mouse monoclonal antibody (Abcam; red, osteocalcin-producing cells and extracellular osteocalcin), DAPI (Roche—Millipore Sigma, Sigma-Aldrich Co, Milan, Italy; blue, nuclei), and phalloidin (Invitrogen–Thermo Fisher Scientific, Waltham, MA, USA; green, cytoplasm) and scanned with a 3-channel epifluorescence microscope (red, green, blue). The analysis was performed on day 21 (in osteogenic medium) in neat hydrogel scaffolds seeded with 2000 cell/mm^3^ hBM-MSCs.

## 5. Conclusions

In summary, we have successfully designed and developed composite hybrid scaffolds with a core-shell structure suitable for bone tissue regeneration by using a novel, benign, and scalable synthesis approach. The developed synthesis strategy begins with melt processing of PLA-based blend/SAP particles that leads to the creation of tunable-porous three-dimensional structures after water treatment, owing to the leaching out of SAP. Subsequent hydrogel grafting allowed us to obtain a core-shell structure in which the core has the main function of temporary mechanical support, whereas the shell enhances biocompatibility and provides better biomimicry for cell culture. More importantly, this production approach allows us to fabricate a wide range of composite scaffolds with different compositions and mechanics that are relevant in guiding tissue regeneration. The PLA-PCL-Hy hybrid scaffolds resulted in an exceptional improvement in mechanical properties compared to the pure hydrogel scaffolds, while maintaining the advantage of surface biocompatibility and bioactivity due to the addition of the shell layer to the core. In vitro analyses have shown excellent results in terms of capability of PLA-PCL-Hy scaffolds to sustain cell viability, proliferation, and osteogenic differentiation. In fact, cells colonized not only the surface but the whole 3D thickness of the scaffold. Moreover, the hydrogel selected for the shell of the scaffolds revealed to be a suitable substrate for bone tissue engineering because differentiation in osteogenic medium resulted in both intracellular and extracellular positive osteocalcin staining, which was indicative of osteoblast differentiation of the human BM-MSCs.

## 6. Patents

Sartore, L.; Russo, D.; Pandini, S.; Nicolai, P.; Ferrari, M.; Gilbert, R.; Irish, J. Integrated core-shell bioactive structure for the regeneration of bone and osteochondral tissues. PCT/IB2021/056113 (8 July 2021).

## Data Availability

Data are contained within the article.

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
