# Peer review of "Hybrid Core-Shell Polymer Scaffold for Bone Tissue Regeneration"

_ijms, 2022, doi:10.3390/ijms23094533_

Round 1

Reviewer 1 Report

The present study has focused on the development of a core-shell bone substitute consisting of porous PLA-PCL as the core and gelatin/chitosan hydrogel as the shell. I like the idea and I feel despite the reporting of main results through a patent, it can be of particular interest. I have noticed some errors and mistakes throughout the paper and also I have got some questions and demands all of which is supposed to improve the quality as follows:

First of all, the manuscript has no conclusion section! The authors numbered each section and 5 is missed! The subcategories also require correction as the numbers in the materials and methods are incorrect.

In the case of TGA and FTIR results, the figures do not have labling of the samples and so how the reader can recognize each pattern of the related sample. Nonetheless, FTIR spectra requires indexing and the main bands should be numbered in the figure.

Although the English is ok, I noticed some errors/misspelling  and I encourage the authors to recheck the whole manuscript.

For instance, in the introduction, using ''consisting in'' is very rare in English writing and also it is used for defining and not mentioning the parts and better to change to consisting of.

Figure 2 reports the results of thermogravimetric analyses (TGA) on (of) PLA-PCL.....

20 ÷ 50 μm particle size

Speaking of surface coating, I am curious to see what is the thickness of the coating. I recommend addition of cross-section SEM images as your samples were reported to be plate-like.

What is the logic behind producing the core with porous structure?

Have you fabricated the core-shell material without using SAP particles as templating agent? As you mentioned the mechanical support of this material comes from the core and so if there was no pore, I suppose the mechanical properties increase, am I right?

As the hydrogel is natural-based, its degradation rate is faster than the core obviously, but there is no degradation study to see how long the coating can endure the physiological solution.

Author Response

Reviewer 1

Comments and Suggestions for Authors

The present study has focused on the development of a core-shell bone substitute consisting of porous PLA-PCL as the core and gelatin/chitosan hydrogel as the shell. I like the idea and I feel despite the reporting of main results through a patent, it can be of particular interest. I have noticed some errors and mistakes throughout the paper and also I have got some questions and demands all of which is supposed to improve the quality as follows:

  • First of all, the manuscript has no conclusion section! The authors numbered each section and 5 is missed! The subcategories also require correction as the numbers in the materials and methods are

Response: The authors thank the reviewer for her/his comments. Conclusions have been added to the manuscript, as reported below, and the numbers of the sections have been corrected.

Line 425: In summary, we have successfully designed and developed composite hybrid scaffolds with a core-shell structure suitable for bone tissue regeneration, using a novel, benign, and scalable synthesis approach. The developed synthesis strategy begins with melt processing of PLA-based blend/SAP particles, that leads to create tunable-porous three-dimensional structures after water treatment, owing to the leaching out of SAP. Subsequent hydrogel grafting allowed to obtain a core-shell structure in which the core has the main function of temporary mechanical support, while the shell enhances biocompatibility and provides better biomimicry for cell culture. More importantly, this production approach allows to fabricate a wide range of composite scaffolds with different compositions and mechanics that are relevant in guiding tissue regeneration. The PLA-PCL-Hy hybrid scaffolds resulted in an exceptional improvement in mechanical properties compared to the pure hydrogel scaffolds, while maintaining the advantage of surface biocompatibility and bioactivity due to the addition of the shell layer to the core. In vitro analyses have shown excellent results in terms of capability of PLA-PCL-Hy scaffolds to sustain cell viability, proliferation, and osteo-genic differentiation. In fact, cells colonized not only the surface but the whole 3D thickness of the scaffold. Moreover, the hydrogel selected for the shell of the scaffolds revealed to be a suitable substrate for bone tissue engineering, since differentiation in osteogenic medium resulted in both intracellular and extracellular positive osteocalcin staining, which was indicative of osteoblast differentiation of the human BM-MSCs.

  • In the case of TGA and FTIR results, the figures do not have labling of the samples and so how the reader can recognize each pattern of the related sample. Nonetheless, FTIR spectra requires indexing and the main bands should be numbered in the figure.

Response: The authors would like to thank the reviewer for her/his recommendations. Unfortunately an error occurred during the upload of the figures. TGA and FTIR graphs have been corrected as kindly suggested and figure 1 and figure 2 have been replaced(in attached file).

  • Although the English is ok, I noticed some errors/misspelling  and I encourage the authors to recheck the whole manuscript.

For instance, in the introduction, using ''consisting in'' is very rare in English writing and also it is used for defining and not mentioning the parts and better to change to consisting of.

Figure 2 reports the results of thermogravimetric analyses (TGA) on (of) PLA-PCL.....

20 ÷ 50 μm particle size

Response: The authors thank the reviewer for noticing this inaccuracy. The authors corrected the misspelling highlighted by the reviewer and rechecked the whole manuscript.

  • Speaking of surface coating, I am curious to see what is the thickness of the coating. I recommend addition of cross-section SEM images as your samples were reported to be plate-like.

What is the logic behind producing the core with porous structure?

Have you fabricated the core-shell material without using SAP particles as templating agent? As you mentioned the mechanical support of this material comes from the core and so if there was no pore, I suppose the mechanical properties increase, am I right?

Response: We acknowledge the reviewer as he/she allowed us to clarify these points. The images in figure 3 are already referred to the cross-sections of the specimens, as the authors reported in the revised caption of the image. The coating thickness cannot be highlighted due to the low amount of grafted hydrogel and the high surface extension of the porous core. Indeed, the hydrogel shell covers the surface of the whole core, including the pores, with a very thin layer that represents a cell-friendly environment. The core was produced with a highly porous structure to enable cells penetration into the whole volume of the final core-shell scaffold as shown in figure 5. The interconnected porosity is also fundamental for gases, nutrients and metabolic wastes exchange. Therefore, even if a non-porous core would increase the mechanical properties, it is not a suitable solution to support cells colonization of the scaffold. The following sentences have been added to the text:

Line 215: In addition, the high magnification micrographs (Fig. 3 e) and f)) show more in detail that also the inner part of the material is characterized by interconnected open porous structure, which is conducive to the infiltration of cells.

Line 219: Furthermore, due to the low amount of grafted hydrogel and the high surface extension of the porous core, it is not possible to highlight the thickness of the coating. The hydrogel shell probably covers the surface of the whole core, including the pores, with a very thin layer that represents a favorable environment for cells.

Line 347: The core was produced with a highly porous structure to enable cells penetration into the whole volume of the final core-shell scaffold. The interconnected porosity is also fundamental for gases, nutrients and metabolic wastes exchange.

  • As the hydrogel is natural-based, its degradation rate is faster than the core obviously, but there is no degradation study to see how long the coating can endure the physiological solution.

Response: Thank you very much for arising this question. Indeed the two components, the core and the shell, have different biodegradation mechanism and kinetics. To elucidate this point the following sentence have been added to the Introduction:

Line 114: The two components also have different degradation timing: the thermoplastic is typically resorbed within 1-2 years [29], therefore retaining mechanical stability for long times, while the hydrogel maintains its integrity up to 1-2 months [10], rapidly creating space for the growth of new bone tissue. 

Reviewer 2 Report

The manuscript is very well written and the results/discussion are well organized. some questions regarding the results are still need to be refined:

  1. Figure 1, 2. Please clearly label the graphs with the meaning of the curves. A more precise figure legend should be considered for better understanding.
  2. It is mentioned in the article that the growth of cells in vitro is more optimistic, but it is doubtful whether the situation in vivo is like that in vitro, can the corresponding data be provided to support it. Also, does the material cause a subsequent inflammatory response after implantation and if so, does it require the addition of appropriate pharmaceutical reagents or the response could be restricted in a certain tolerable range.
  3. In Figure 3, whether the leaching degree of SAP can be illustrated with more data/description; whether the residual SAP will have an impact on the subsequent experiments and readout.
  4. As a bone filler, can it coexist well with one's own bone after the bone regeneration is completed? Or can it gradually disappear through metabolism/ degradation-resorption?
  5. It is proposed in the abstract that adequate stimulation can lead to differentiation of cells toward an osteogenic phenotype. Can in vitro experiments be performed to test this conjecture or this is just speculation.

Author Response

Reviewer 2

Comments and Suggestions for Authors

The manuscript is very well written and the results/discussion are well organized. some questions regarding the results are still need to be refined:

  1. Figure 1, 2. Please clearly label the graphs with the meaning of the curves. A more precise figure legend should be considered for better understanding.

Response: The authors thank the reviewer for her/his comments. Unfortunately an error occurred during the upload of the figures. TGA and FTIR graphs have been corrected as kindly suggested and figure 1 and 2 have been replaced as reported above (Reviewer 1, response 2).

  1. It is mentioned in the article that the growth of cells in vitro is more optimistic, but it is doubtful whether the situation in vivo is like that in vitro, can the corresponding data be provided to support it. Also, does the material cause a subsequent inflammatory response after implantation and if so, does it require the addition of appropriate pharmaceutical reagents or the response could be restricted in a certain tolerable range.

Response: The authors thank the reviewer for her/his comments. Preliminary in vivo and ex vivo studies showed encouraging results in this respect, and they will be published in the near future. The following sentence has been added to enrich the Discussion section:

Line 418: Preliminary in vivo studies on mandibular reconstruction in a rabbit model [40] also found the scaffolds to be biocompatible, as they were well tolerated without any anti-inflammatory treatment. No signs of surgical site infection were observed after implantation. Besides, the scaffolds were well integrated with the bone tissue surrounding the mandibular defect as well as with the new growing tissue, which was found to gradually replace them, as observed during ex vivo histochemical analyses.

  1. In Figure 3, whether the leaching degree of SAP can be illustrated with more data/description; whether the residual SAP will have an impact on the subsequent experiments and readout.

Response: We thank the reviewer for arising this question. We added the following paragraph to the Results to better describe the complete particle leaching:

Line 195: During the water treatment the SAP particles, probably due to the low interaction with the matrix, swell and leach out from the matrix generating high porosity. As demonstrated in a previous study [20], particles leaching was completed after immersion in water for 15 days. Indeed, the elemental analysis carried out during the SEM observation before the water treatment confirmed the identification of SAP particles. On the other hand, after the water treatment, the same analysis carried out in different points of the sample, did not detect the presence of the SAP components, suggesting the correct removal of the SAP particles.

  1. As a bone filler, can it coexist well with one's own bone after the bone regeneration is completed? Or can it gradually disappear through metabolism/ degradation-resorption?

Response: We thank the reviewer for arising this question. We added the following paragraph to the Introduction and Discussion sections to clarify this point:

Line 114: The two components also have different degradation timing: the thermoplastic is typically resorbed within 1-2 years [29], therefore retaining mechanical stability for long times, while the hydrogel maintains its integrity up to 1-2 months [10], rapidly creating space for the growth of new bone tissue.

Line 418: Preliminary in vivo studies on mandibular reconstruction in a rabbit model [40] also found the scaffolds to be biocompatible, as they were well tolerated without any anti-inflammatory treatment. No signs of surgical site infection were observed after implantation. Besides, the scaffolds were well integrated with the bone tissue surrounding the mandibular defect as well as with the new growing tissue, which was found to gradually replace them, as observed during ex vivo histochemical analyses.

  1. It is proposed in the abstract that adequate stimulation can lead to differentiation of cells toward an osteogenic phenotype. Can in vitro experiments be performed to test this conjecture or this is just speculation.

Response: We thank the reviewer for this comment. We have conducted in vitro differentiation of human bone marrow-derived mesenchymal stromal cells (hBM-MSCs) for 21 days in osteogenic medium containing L-Ascorbic acid, β-glycerophosphate and Dexamethasone. hBM-MSCs were stained for osteocalcin (a mature osteoblast maker) and counter-stained for Phalloidin-DAPI, as noted in Figure 5C. Additional text to this effect has been added in section 2.4, and in section 4.4:

Line 282: To demonstrate that the hBM-MSCs were capable of osteogenic differentiation on the hydrogels, we performed a standard 3-week osteogenic differentiation assay. Figure 5C,D reports epifluorescence microscopy of the neat hydrogel seeded with 2000 cell/mm3 hBM-MSCs, on which cells were capable of proliferating (Figure 5D) and differentiating towards the osteogenic phenotype (Figure 5C), as evidenced by positive staining for osteocalcin, a mature osteoblast bone formation biomarker.

P18 L618: (or day 25 post-seeding)

P18 L623: (in osteogenic medium)

Round 2

Reviewer 1 Report

The authors have addressed my comments satisfactorily.